

# H₂SO₄-H₂O binary and H₂SO₄-H₂O-NH₃ ternary homogeneous and ion-mediated nucleation: Lookup tables for 3-D modeling application

Fangqun Yu[1], Alexey B. Nadykto[1, 2, 3], Gan Luo[1], and Jason Herb[1]

Correspondence to: F. Yu (fyu@albany.edu)

[1] Atmospheric Sciences Research Center, University at Albany, Albany, New York, US

[2] Department of Applied Mathematics, Moscow State University of Technology "Stankin", Russia

[3] National Research Nuclear University MEPhI (Moscow Engineering Physics Institute), Department of General Physics, Russian Federation

**Abstract**. Formation of new particles in the atmosphere have important implications for air quality and climate. Recently, we have developed a kinetically-based H₂SO₄-H₂O-NH₃-Ions nucleation model which captures well the absolute values of nucleation rates as well as dependencies of nucleation rates on NH₃ and H₂SO₄ concentrations, ionization rates, temperature, and relative humidity observed in the well-controlled Cosmics Leaving OUtdoor Droplets (CLOUD) measurements. Here we employ the aforementioned recently developed kinetic nucleation model to generate nucleation rate look-up tables for H₂SO₄-H₂O binary homogenous nucleation (BHN), H₂SO₄-H₂O-NH₃ ternary homogeneous nucleation (THN), H₂SO₄-H₂O-Ions binary ion-mediated nucleation (BIMN), and H₂SO₄-H₂O-NH₃-Ions ternary ion-mediated nucleation (TIMN). A comparison of nucleation rates calculated using the lookup tables with CLOUD measurements of BHN, BIMN, THN, and TIMN is presented. The lookup tables cover a wide range of key parameters controlling binary, ternary and ion-mediated nucleation in the Earth's atmosphere, and are a cost-efficient solution for multi-dimensional modeling. The lookup tables and FORTRAN codes, made available through this work, can be readily used for BHN, THN, BIMN, and TIMN simulations in 3-D modeling. The lookup tables can also be used by experimentalists involved in laboratory and field measurements for a quick assessment of nucleation involving H₂SO₄, H₂O, NH₃, and Ions.

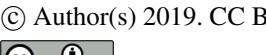



## 1. Introduction

Particles in the troposphere either come from direct emission (i.e., primary particles) or in-situ nucleation (i.e., secondary particles). Secondary particles formed via nucleation dominate the number concentrations of atmospheric particles (Spracklen et al., 2008; Pierce and Adams, 2009;
Yu and Luo, 2009) that are important for air quality and climate. Nucleation in the atmosphere is a dynamic process involving various interactions of precursor gas molecules, small clusters, and pre-existing particles (Yu and Turco, 2001; Zhang et al., 2012; Lee et al., 2019). $H_2SO_4$ and $H_2O$ are known to play an important role in atmospheric new particle formation (NPF) (e.g., Doyle, 1961). It has been long known that while binary homogeneous nucleation (BHN) of $H_2SO_4$-$H_2O$
may play a dominant role in the cold upper troposphere, it cannot explain nucleation events observed in the lower troposphere (e.g., Weber et al., 1996). Two alternative nucleation theories have been proposed, one is the ternary homogeneous nucleation (THN) involving $NH_3$ (Coffman and Hegg, 1995; Napari et al., 2002) and the other is the ion-mediated nucleation (IMN) considering the role of the ubiquitous ions in enhancing the stability and growth of pre-nucleation
clusters (Yu and Turco, 2001). The laboratory measurements in the CLOUD (Cosmics Leaving Outdoor Droplets) chamber experiments at CERN show that both ammonia and ionization can enhance $H_2SO_4$-$H_2O$ nucleation (Kirkby et al., 2011). In order to reach a deep and insightful understanding of the physico-chemical processes underlying the observed enhancement effect of ammonia and ions, Yu et al. (2018) developed a kinetic ternary ion-mediated nucleation (TIMN)
model for $H_2SO_4$-$H_2O$-$NH_3$-Ion system, with thermodynamic data derived from laboratory measurements and quantum-chemical calculations. The model is able to explain the observed difference in the effect of $NH_3$ in lowering the nucleation barriers for clusters of different charging states and predicts nucleation rates in good agreement with CLOUD observations (Yu et al., 2018).

The main objective of this work is to employ the recently developed kinetic nucleation model
(Yu et al., 2018) to generate nucleation rate look-up tables for four different nucleation pathways: $H_2SO_4$-$H_2O$ binary homogenous nucleation (BHN), $H_2SO_4$-$H_2O$-$NH_3$ ternary homogeneous nucleation (THN), $H_2SO_4$-$H_2O$-Ions binary ion-mediated nucleation (BIMN), and $H_2SO_4$-$H_2O$-$NH_3$-Ions ternary ion-mediated nucleation (TIMN). With the look-up tables and simple interpolation subroutines, the computational costs of the binary and ternary nucleation rate
calculations were significantly reduced that is critically important for multi-dimensional modeling.



The computed nucleation rates of BHN, THN, BIMN, and TIMN based on the look-up tables were evaluated against CLOUD measurements.

## 2. Nucleation rate lookup tables for BHN, THN, BIMN, and TIMN

The $H_2SO_4$-$H_2O$-$NH_3$-Ions kinetic nucleation model, as described in detail in Yu et al. (2018), solves the dynamic interactions of various clusters and offers a physics-based explanation of the different concentrations of $NH_3$ needed to induce nucleation on neutrals, positive ion, and negative ions. The model is designed for a nucleating system consisting of $H_2SO_4$-$H_2O$-$NH_3$ in t the presence of ionization (i.e, ternary ion-mediated nucleation, TIMN). In the absence of $NH_3$, the model transforms into the binary homogeneous nucleation (BHN) or binary ion-mediated nucleation (BIMN) and reduces to BHN or ternary homogeneous nucleation (THN) in the case when no ions are present. It is important to note that in the $H_2SO_4$-$H_2O$-$NH_3$ ternary system, binary $H_2SO_4$-$H_2O$ clusters co-exist with ternary $H_2SO_4$-$H_2O$-$NH_3$ ones while in the system with ions, neutral clusters co-exist with charged clusters. Therefore, BIMN includes BHN, THN includes BHN, and TIMN includes both BIMN and THN.

For the benefit of different applications and for enabling one to evaluate the contribution of different nucleation pathways (binary versus ternary, neural versus ion-mediated), we run the model to generate nucleation lookup tables separately for the four different nucleating systems, i.e, $H_2SO_4$-$H_2O$ (BHN), $H_2SO_4$-$H_2O$-$NH_3$ (THN), $H_2SO_4$-$H_2O$-Ions (BIMN), and $H_2SO_4$-$H_2O$-$NH_3$-Ions (TIMN). One can accurately determine the role of $NH_3$ by looking into the difference between BHN (BIMN) and THN (TIMN) rates, and the role of ionization by examining the difference between BHN (THN) and BIMN (TIMN) rates. Another benefit of generating separate lookup tables is that for the users who are only interested in BHN, BIMN, or THN, the corresponding lookup tables are much smaller than that of TIMN) and much easier to handle.

For many practical applications, the steady state nucleation rates under given conditions are required. Nucleation rates are conventionally calculated at the sizes of critical clusters (Seinfeld and Pandis, 2016). Since the kinetic nucleation model explicitly solves the evolution of clusters of various sizes, it can calculate steady state particle formation rates at any sizes larger than critical sizes (Yu, 2006). In many laboratory studies new particle formation rates were measured at certain detection sizes, typically much larger than critical sizes. For example, the nucleation rates measured at CLOUD are for particles with mobility diameter of 1.7 nm. For the atmospheric



modeling with size resolved particle microphysics, the sizes of the first bin are generally much larger than the critical sizes and the nucleation rates calculated at the critical sizes (which vary with the atmospheric conditions) have to be extrapolated to the sizes of the first bin based on the assumed growth rates and coagulation sinks of freshly nucleated particles that may lead to

additional uncertainties. To compare model nucleation rates with typical laboratory measurements and to facilitate the application of the obtained results in size-resolved particle microphysics model that can have first bin down to the size of around 1 – 2 nm, nucleation rates are calculated at 1.7 nm mobility diameter (corresponding to mass diameter of ~ 1.5 nm) (Yu et al., 2018).

Look-up table of steady state nucleation rates for BHN ($J_{BHN}$), THN ($J_{THN}$), BIMN ($J_{BIMN}$), and

TIMN ($J_{TIMN}$) have been generated under a wide range of atmospheric conditions. There are six parameters controlling $J_{TIMN}$: sulfuric acid vapor concentration ($[H_2SO_4]$), ammonia gas concentration ($[NH_3]$), T, RH, ionization rate (Q), and surface area of pre-existing particles (S). Compared to $J_{TIMN}$, there is one less controlling parameter for both $J_{BIMN}$ (no $[NH_3]$ dependence) and $J_{THN}$ (no Q dependence) while $J_{BHN}$ only depends on four parameters ($[H_2SO_4]$ T, RH, and S).

Table 1 gives the range of each dependent variable dimension, total number of points in each dimension, values at each point, and controlling parameters for the four nucleation pathways. The range and resolution in each parameter space is designed based on the sensitivity of nucleation rates to the parameter, its possible range in the troposphere, and a balance between the accuracy and sizes of the lookup tables. T covers from 190 K to 304 K (resolution: 3 K) and RH from 0.5%

to 99.5% (resolution: 4%). For $[H_2SO_4]$, we use 31 points to cover $5\times10^5$ to $5\times10^8$ cm$^{-3}$ plus one additional point at $[H_2SO_4]= 5\times10^9$ cm$^{-3}$. For $[NH_3]$, we employ 31 points to cover $10^8$ to $10^{11}$ cm$^{-3}$ plus two additional points at $[NH_3]= 10^5$ cm$^{-3}$ and $10^{12}$ cm$^{-3}$. Q ranges from 2 – 23 ion-pairs cm$^{-3}$s$^{-1}$ with the resolution of 5 values per decade (geometric) plus one additional point at Q =100 ion-pairs cm$^{-3}$s$^{-1}$, and S ranges from 20 – 200 $\mu$m$^2$cm$^{-3}$ with two points. Almost all the possible

tropospheric conditions relevant to NPF shall be covered with the above parameter ranges. The look-up table for $J_{TIMN}$ is the largest, being composed of $J_{TIMN}$ at more than 17 million points (32x33x39x26x8x2=17,132,544) and with the total size in the text format of ~103 MB. For comparison, the smallest lookup table (for $J_{BHN}$) has just 64,896 points with total size of ~ 0.38 MB in the text format. For any given values of $[H_2SO_4]$, $[NH_3]$, T, RH, Q, and S within the ranges

specified in Table 1, nucleation rates can be obtained using the look-up tables with an efficient multiple-variable interpolation scheme as described in Yu (2010). For conditions out of the ranges





specified in Table 1, which may occur occasionally in the atmosphere, extrapolation is allowed. The $J_{BHN}$, $J_{THN}$, $J_{BIMN}$, and $J_{TIMN}$ look-up tables can be accessed via the information given in the data availability section and can be used to calculate nucleation rates efficiently in 3-dimensional models.

## 3. Comparison of BHN, THN, BIMN, and TIMN rates from the lookup tables with CLOUD measurements

Dunne et al. (2016) reported CLOUD measured nucleation rates under 377 different conditions
(Table S1 of Dunne et al. (2016)). These data can be divided into BHN, THN, BIMN, and TIMN based on the values of $[NH_3]$ and Q in the chamber. Nucleation is classified as neutral (BHN or THN), when Q=0, and as binary (BHN or BIMN) when $[NH_3] < 0.1$ ppt. As a result, 15, 27, 110, and 225 of these CLOUD measurements correspond to BHN, BIMN, THN, and TIMN, respectively. Figures 1-4 present the comparisons of the nucleation rates calculated from the
lookup tables ($J_{model}$) with corresponding values observed during CLOUD experiments ($J_{obs}$) under BHN, BIMN, THN, and TIMN conditions. The error bars give the $J_{model}$ range as a result of the measured $[H_2SO_4]$ uncertainty (-50%, +100%). The uncertainties in $J_{obs}$ (overall a factor of two) and those associated with the uncertainty in measured $[NH_3]$ (-50%, +100%) are not included.

Because of the increase in the contamination (both unwanted ammonia and amines) with the
CLOUD chamber temperate (Kurten et al., 2016), binary nucleation measurements (i.e., without ammonia, $[NH_3] < 0.1$ ppt) are only available at very lower T (Figs. 1-2). Both BHN and BIMN predictions based on the lookup tables overall agree well with the available CLOUD observations within the uncertainty range. As pointed out earlier, binary $H_2SO_4$-$H_2O$ clusters co-exist with ternary $H_2SO_4$-$H_2O$-$NH_3$ ones in the ternary system while neutral clusters co-exist with charged
clusters in the system containing ions. Therefore, the nice agreement of BHN and BIMN model predictions with observations provides good foundation for the more complex THN and TIMN models.  CLOUD experiments have more data points for THN and TIMN, under a wide temperature range covering the lower troposphere. For the THN (Fig. 3), the model prediction is consistent with measurements at low temperature (T= ~ 205 – 250 K) but deviates from
measurements at high T, with the level of model under-prediction increasing with increasing T. As pointed out in Yu et al., (2018), the level of contamination in the CLOUD chamber appears to





increase with temperature (Kurten et al., 2016), the nice agreement at lower T and the deviation at higher T may be associated with contaminations (such as amines, etc.) in the CLOUD (Kirkby et al., 2011) which increase with temperature (Kurten et al., 2016). In contrast to THN, $J_{model}$ for TIMN (Fig. 4) agrees with CLOUD measurements within the uncertainties under nearly all

5 conditions. $J_{model}$ for TIMN at T=292 – 300 K is slightly lower than the corresponding observed values, likely a result of similar reasons for the THN under-prediction at higher T (Fig. 3). As demonstrated in Yu et al. (2018), the nucleation of ions is typically stronger than that of neutral clusters for both binary and ternary nucleating systems. The ubiquitous presence of ionization in the Earth's atmosphere calls for regional and global aerosol models to take into account the effect

of ionization in NPF. The BIMN and TIMN lookup tables, derived from a physics-based kinetic nucleation model and validated against the state-of-the-art CLOUD measurements, provide an efficient way to incorporate the role of ionization in new particle formation in 3-D models.

**Acknowledgments. We** acknowledge funding support from the US National Science Foundation

(grant #: AGS1550816), the Russian Science Foundation, and the Ministry of Science and Education of Russia (grants #: 1.6198.2017/6.7 and 1.7706.2017/8.9).

**Code and data availability.** The code and lookup tables can be accessed via the zenodo data repository http://doi.org/10.5281/zenodo.3483797. For quick calculation of BHN, THN, BIMN,

and TIMN rates under specified conditions, one can use the online nucleation calculators which we have developed based on these lookup tables and made available to the public at http://apm.asrc.albany.edu/nrc/.

**Author Contribution**. FY designed the lookup tables and generated the lookup tables. FY, AN,

GL, and JH contributed to the kinetic model used to generate the lookup tables. FY wrote the paper with contributions from all co-authors.

**Competing interests**. The authors declare that they have no conflict of interest.

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





**Table 1**. The range of values for each independent variable in the BHN, THN, BIMN, and TIMN nucleation rate lookup tables. Also given are the total number of values for each variable, the specific values at which nucleation rates have been calculated, and controlling parameters for the four nucleation mechanisms.

| Parameters | Range | Total # of points | Values at each point | Controlling Parameters | | | |
|---|---|---|---|---|---|---|---|
| | | | | BHN | THN | BIMN | TIMN |
| $[H_2SO_4]$ $(cm^{-3})$ | $5 \times 10^5 - 5 \times 10^9$ | 32 | $[H_2SO_4](i)=5 \times 10^5 \times 10^{(i-1)/10}$, $i = 1, 31$ $[H_2SO_4](32)=5 \times 10^9$ | x | x | x | x |
| T (K) | 190 – 304 | 39 | $T(j)=190 + 3 \times (j-1), j =1, 39$ | x | x | x | x |
| RH (%) | 0.5 – 99.5 | 26 | $RH(1)=0.5, RH(k)=4 \times (k-1)$, $k=2, 25; RH(26)= 99.5$ | x | x | x | x |
| S $(\mu m^2 cm^{-3})$ | 20 – 200 | 2 | $S(1)= 20, S(2)= 200$ | x | x | x | x |
| $[NH_3]$ $(cm^{-3})$ | $10^5 - 10^{12}$ | 33 | $[NH_3](1)=10^5, [NH_3](m)= 10^8 \times 10^{(m-1)/10}, m = 2, 32$; $[NH_3](33) =10^{12}$ | | x | | x |
| Q (ion-pairs $cm^{-3}s^{-1}$) | 2 – 100 | 8 | $Q(n)= 2 \times 1.5^{(n-1)}, n = 1, 7$; $Q(8)=100$ | | | x | x |



**Figures**

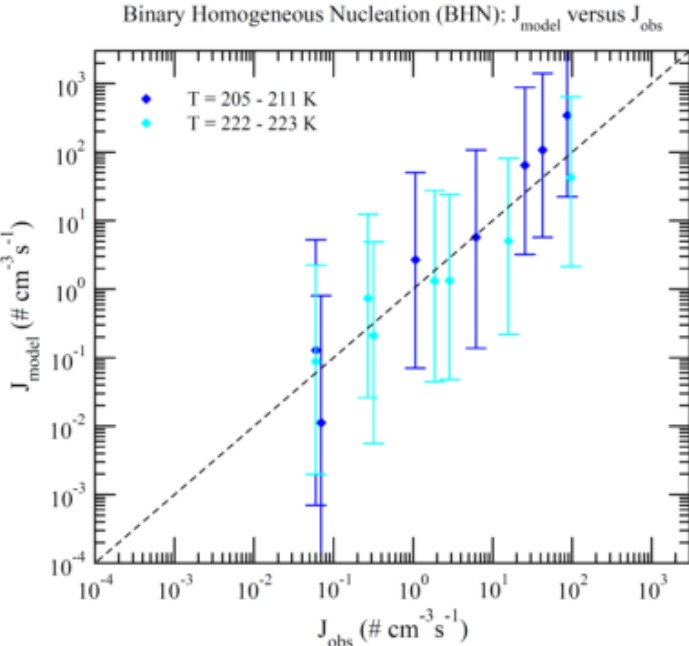

5     Figure 1. Model predicted ($J_{model}$) versus observed ($J_{obs}$) nucleation rates under BHN conditions (no ionization, [$NH_3$]<0.1 ppt) of CLOUD measurements reported in Table S1 of Dunne et al. (2016). The data points are grouped according to temperatures as specified in the legend. Vertical error bars show the range of $J_{model}$ calculated at 50% and 200% of measured [$H_2SO_4$], corresponding to the uncertainties in measured [$H_2SO_4$] (-50%, +100%). Error bars associated with

10    the uncertainties in measured [$NH_3$] (-50%, +100%), and $J_{obs}$ (overall a factor of two) are not shown.



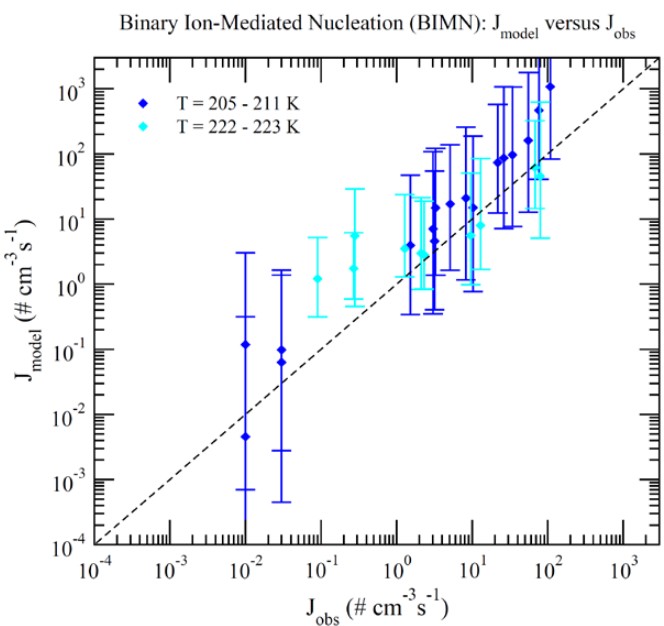

Figure 2. Same as Figure 1 but for BIMN conditions (with ionization, [NH$_3$]<0.1 ppt).

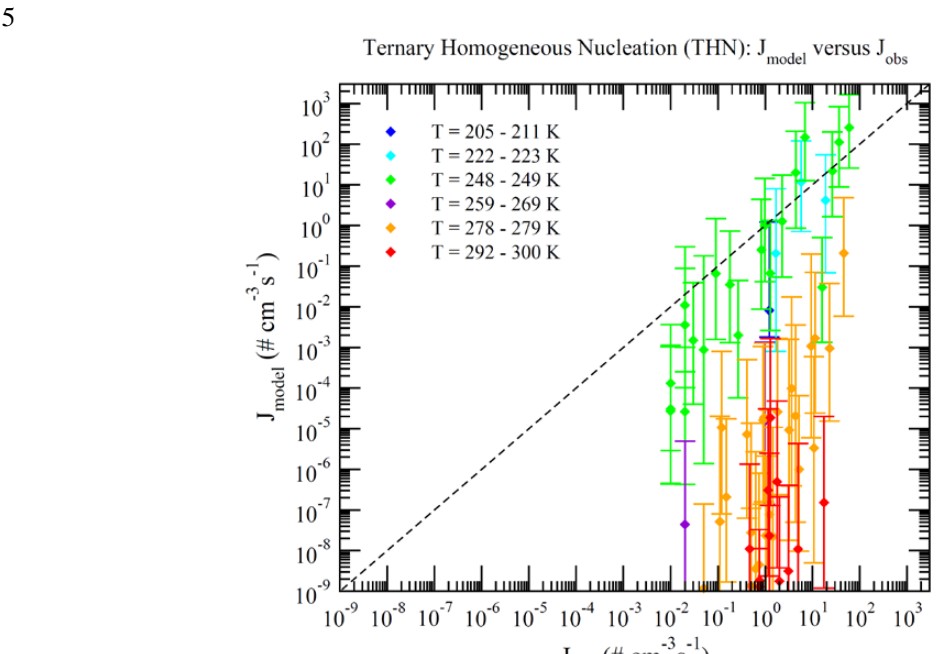

Figure 3. Same as Figure 1 but for THN conditions (no ionization, [NH$_3$]>0.1 ppt).



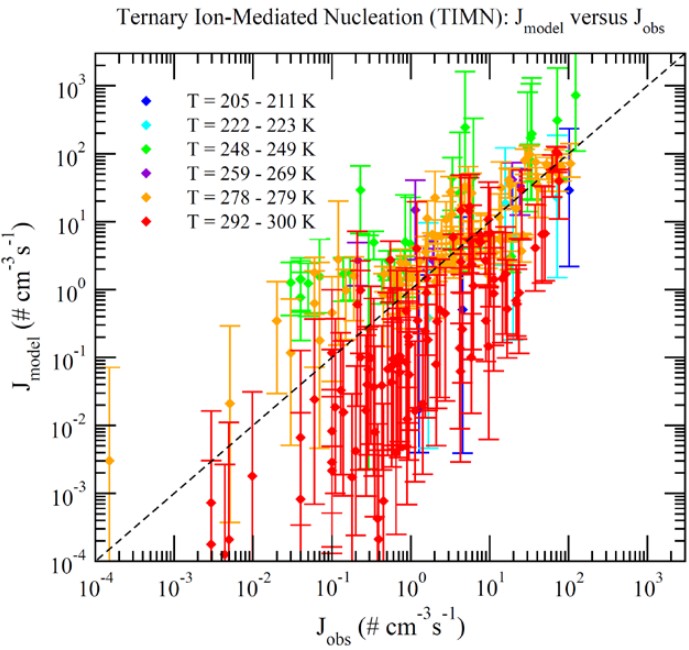

Figure 4. Same as Figure 1 but for TIMN conditions (with ionization, [NH$_3$]>0.1 ppt).