# Peer review of "H2SO4-H2O binary and H2SO4-H2O-NH3 ternary homogeneous and ion-mediated"

_Geoscientific Model Development, 2019_

## Short Comment (SC1) · 4 Dec 2019

Dear authors,

please consider, that the loopuk tables might be changed in the future. Therefore please provide a version number of the tables in the title of the manuscript.

Best regards, Astrid Kerkweg (Executive Editor)
* * *

---

## Referee Comment (RC1) · Anonymous Referee #1 · 2 Jan 2020

This work documents aerosol-nucleation-rate lookup-tables generated based on the results from a kinetic model (Yu et al. 2018ACP). Four major aerosol nucleation mechanisms are considered in these lookup-tables and they cover a wide range of key parameters relevant for aerosol nucleation. The lookup-tables can be used in 3-D models to save computational cost, so they are potentially useful for other modelers who want to simply the aerosol nucleation treatment in their models.

Overall, the manuscript is well written. The documentation is clear and key information is provided. I think the work could be further improved by comparing these tables (calculated nucleation rates) with other widely-used aerosol nucleation parameterizations,

so that other users can have an idea about what to expect in their model result. For example, the Vehkamaki et al (2002, hereafter V2002) binary nucleation scheme is used by many aerosol models (e.g. CAM5 Liu et al., 2012GMD, ECHAM5-HAM, Stier et al. 2005ACP, etc). How does the BHN lookup-table compare with V2002? The ion-induced aerosol nucleation is considered in the ECHAM5-HAM2 model (Zhang et al., 2012ACP), using a similar lookup-table method (Kazil and Lovejoy ,2007ChemPhys and Kazil et al, 2010ACP). How does the BIMN lookup-table compare with K2010?

In addition, some aerosol models (Wang et al, 2009ACP, Zhang et al, 2012) use the nucleation parameterization for the boundary layer (e.g. Kuang et al., 2008JGR) in combination with the binary nucleation scheme. Can the THN/BIMN/TIMN schemes be used along with such boundary nucleation scheme? It would be nice to provide such information to other users as well.

Other specific/minor comments:

P1L11, abstract: have -> has

P1L12, L30 and throughout the text: Better use either "Ion" or "ion".

P1L25: "for BHN, THN, BIMN, and TIMN" could be deleted

P2L11: Maybe also mention the nucleation pathways involving organics?

P4L12: Is RH the hybrid relative humidity or the RH respect to water? Please clarify.

P4L15-24: It would be nice to provide some quantitative estimate of the lookup-table accuracy here.

P4L24: Can two points for S to get sufficient accuracy?

P5L1: extrapolation -> linear?

P5L21: very lower -> very low

P6L20: the online program (http://apm.asrc.albany.edu/nrc/) didn't work for me (both in

safari and firefox). Better fix it before the final publication.

Reference

Kazil, J., Stier, P., Zhang, K., Quaas, J., Kinne, S., O'Donnell, D., Rast, S., Esch, M., Ferrachat, S., Lohmann, U., and Feichter, J.: Aerosol nucleation and its role for clouds and Earth's radiative forcing in the aerosol-climate model ECHAM5-HAM, Atmos. Chem. Phys., 10, 10733–10752, https://doi.org/10.5194/acp-10-10733-2010, 2010.

Kuang, C., McMurry, P.H., McCormick, A.V. and Eisele, F.L., 2008. Dependence of nucleation rates on sulfuric acid vapor concentration in diverse atmospheric locations. Journal of Geophysical Research: Atmospheres, 113(D10).

Vehkamäki, H., Kulmala, M., Napari, I., Lehtinen, K.E., Timmreck, C., Noppel, M. and Laaksonen, A., 2002. An improved parameterization for sulfuric acid–water nucleation rates for tropospheric and stratospheric conditions. Journal of Geophysical Research: Atmospheres, 107(D22), pp.AAC-3.

Stier, P., Feichter, J., Kinne, S., Kloster, S., Vignati, E., Wilson, J., Ganzeveld, L., Tegen, I., Werner, M., Balkanski, Y., Schulz, M., Boucher, O., Minikin, A., and Petzold, A.: The aerosol-climate model ECHAM5-HAM, Atmos. Chem. Phys., 5, 1125–1156, https://doi.org/10.5194/acp-5-1125-2005, 2005.

Zhang, K., O'Donnell, D., Kazil, J., Stier, P., Kinne, S., Lohmann, U., Ferrachat, S., Croft, B., Quaas, J., Wan, H., Rast, S., and Feichter, J.: The global aerosol-climate model ECHAM-HAM, version 2: sensitivity to improvements in process representations, Atmos. Chem. Phys., 12, 8911–8949, https://doi.org/10.5194/acp-12-8911-2012, 2012.
* * *

---

## Referee Comment (RC2) · Anonymous Referee #2 · 3 Feb 2020

F. Yu and co-workers have developed lookup tables for quickly and efficiently obtaining new-particle formation ("nucleation") rates based on Yu's 2018 model, which includes H2SO4, NH3, H2O and ions. This is a very useful tool for researchers in the atmospheric aerosol field. Since this manuscript doesn't deal with the nucleation model as such (it's taken as a given), I won't comment on any of the potential issues with the model itself, but only on the application described here. However I will note that I fully agree with the executive editor's request of providing version numbers - while Yu 2018 is an impressive model, it is unlikely to be *perfect* in the sense that no further improvement would ever be possible.

Some minor issues to take in to account when preparing the final manuscript:

-The authors say that their rates can be compared with typical laboratory measurements. How should wall losses be accounted for in the comparison - is the idea that users should just scale the "S" parameter in the model to roughly fit the losses in the experiment? How well does this actually capture the effect of wall losses (especially in e.g. flow-tube experiments?).

-The authors say that Q ranges from 2 to 23 ion pairs / cm3; for the benefit of casual readers just skimming the text they might mention here that the Q=0 case is also covered (as they actually have separate look-up tables for this case).

-"extrapolation is allowed": this sweeping statement sounds potentially a bit dangerous; have the authors actually tested how well extrapolations work? Perhaps give the readers some guidelines on what kind of extrapolations are recommended, and/or some caveats as to when they can be expected to work (and when not)?

-There seems to be an extra bracket in TIMN) on line page 3, 24

―――――――――――――――

---

## Referee Comment (RC3) · Anonymous Referee #3 · 4 Feb 2020

This paper describes look-up tables to speed up the implementation of the state-of-the-art Yu et al (2018) ternary nucleation parameterization in atmospheric models. These look-up tables have the potential to be very helpful to atmospheric modelers and they are well described both in the manuscript and in the well-documented code available via Zenodo. I recommend this paper for publication. I have only minor improvements to suggest.

I note that I was able to use the online program successfully, despite the comment of one of the other reviewers, so I can comment that the authors have presumably fixed it.

The parameterization documented by Yu et al (2018) has not been, as far as I am aware, tested under all atmospheric or planetary science conditions, and so I think its range of validity could be discussed in this paper a little more thoroughly to ensure atmospheric modelers are aware of its possible limitations.

In polluted conditions or where there are high concentrations of biogenic organic molecules, I think it is possible that the HSO4- ion concentrations predicted by the model for a given ion production rate could be biased high since other molecules may be ionized instead. I appreciate that in these conditions the ions will be mostly lost to high condensation sinks, and so nucleation is likely to be dominated by neutral processes. So it is unlikely to be a big effect, but still perhaps worth mentioning.

Similarly, the comment that 'extrapolation is allowed' for conditions out of range of the table might need qualifying, since nucleation rates are very non-linear. While I appreciate that extrapolation from this parameterization should be more robust than extrapolation from, for example, the empirical parameterization of Dunne et al (2016), it still necessarily leads to uncertainty. In particular, I think nucleation rates at very low relative humidity (below 0.5%) or at temperatures above 300K are still very uncertain and extrapolating from the tables may lead to errors. Could the binary part of the parameterization be used for nucleation on Venus, for example, as discussed by Määttanen et al (JGR 2018; https://doi.org/10.1002/2017JD027429), or in the stratosphere?

As the authors address the comment of reviewer #1 on comparison to other parameterizations, the Määttanen et al paper should be discussed as it is in some respects an update of Vehkamaki et al (2002).

On page 3, line 8, there is an extraneous 't'. On page 6, line 8, I think it's worth specifying "ternary nucleating systems with ammonia" because while the statement is perfectly correct for the ammonia system, ternary nucleation of other molecules (some amines, for example) with sulfuric acid is dominated by neutral processes.

[Figure]

2019.

---

## Author Comment (AC2) · 16 Mar 2020

The authors would like to thank the reviewer for the constructive comments. Our replies to the comments are given below, with the original comments in black, and our response in blue.

**Anonymous Referee #1**

This work documents aerosol-nucleation-rate lookup-tables generated based on the results from a kinetic model (Yu et al. 2018ACP). Four major aerosol nucleation mechanisms are considered in these lookup-tables and they cover a wide range of key parameters relevant for aerosol nucleation. The lookup-tables can be used in 3-D models to save computational cost, so they are potentially useful for other modelers who want to simply the aerosol nucleation treatment in their models.
Overall, the manuscript is well written. The documentation is clear and key information is provided.
Thank you for positive comments affirming potential usefulness of the work.

I think the work could be further improved by comparing these tables (calculated nucleation rates) with other widely-used aerosol nucleation parameterizations, so that other users can have an idea about what to expect in their model result. For example, the Vehkamaki et al (2002, hereafter V2002) binary nucleation scheme is used by many aerosol models (e.g. CAM5 Liu et al., 2012GMD, ECHAM5-HAM, Stier et al. 2005ACP, etc). How does the BHN lookup-table compare with V2002? The ioninduced aerosol nucleation is considered in the ECHAM5-HAM2 model (Zhang et al., 2012ACP), using a similar lookup-table method (Kazil and Lovejoy ,2007ChemPhys and Kazil et al, 2010ACP). How does the BIMN lookup-table compare with K2010? In addition, some aerosol models (Wang et al, 2009ACP, Zhang et al, 2012) use the nucleation parameterization for the boundary layer (e.g. Kuang et al., 2008JGR) in combination with the binary nucleation scheme. Can the THN/BIMN/TIMN schemes be used along with such boundary nucleation scheme? It would be nice to provide such information to other users as well.

This is a good suggestion. We have added discussions on the comparison of present nucleation schemes with other widely-used aerosol nucleation parameterizations.

We think that the THN/BIMN/TIMN schemes shall not be used along with empirical boundary nucleation schemes because these empirical parameterizations were derived from in-situ measurements and might be some kind of simplified parameterizations for THN/BIMN/TIMN processes in the boundary layer. To use both may lead to double count.

Other specific/minor comments:
P1L11, abstract: have -> has
Corrected.

P1L12, L30 and throughout the text: Better use either "Ion" or "ion".
Done.

P1L25: "for BHN, THN, BIMN, and TIMN" could be deleted
Done.

P2L11: Maybe also mention the nucleation pathways involving organics?

Yes. It is now mentioned.

P4L12: Is RH the hybrid relative humidity or the RH respect to water? Please clarify.
It is the RH respect to water.  Clarified in the text describing Table 1.

P4L15-24: It would be nice to provide some quantitative estimate of the lookup-table accuracy here.
Compared to those based on the full model, the deviation of nucleation rates based on the lookup tables is generally within a factor of two, well within the corresponding uncertainty of CLOUD measurements.  We added a discussion on this to the text.

P4L24: Can two points for S to get sufficient accuracy?
The dependence of nucleation rates on the surface area, which serves as coagulation sink of pre-nucleation clusters, is relatively linear and two points for S provide reasonable accuracy (compared to the uncertainties in the model itself and measurements). In the atmosphere, the surface area of pre-existing particles not only serves as coagulation sink but also as condensation sink for $H_2SO_4$, and thus has a more profound impact because nucleation rates are highly sensitive to $[H_2SO_4]$. For the lookup tables, $[H_2SO_4]$ is fixed and therefore the dependence of nucleation rates on surface area are relatively weaker. It should be noted that most of existing nucleation parameterizations do not take into account the effect of surface area. We have clarified this in the text.

P5L1: extrapolation -> linear?

We use linear extrapolation with regard to the dependence of $Log_{10}J$ on surface area. We have clarified this in the text.

P5L21: very lower -> very low
Corrected.

P6L20: the online program (http://apm.asrc.albany.edu/nrc/) didn't work for me (both in safari and firefox). Better fix it before the final publication.
We found that sometime the calculators didn't restart automatically when the server was rebooted. This problem has now resolved and the calculators shall be online all time.

Reference
Kazil, J., Stier, P., Zhang, K., Quaas, J., Kinne, S., O'Donnell, D., Rast, S., Esch, M., Ferrachat, S., Lohmann, U., and Feichter, J.: Aerosol nucleation and its role for clouds and Earth's radiative forcing in the aerosol-climate model ECHAM5-HAM, Atmos. Chem. Phys., 10, 10733–10752, https://doi.org/10.5194/acp-10-10733-2010, 2010.
Kuang, C., McMurry, P.H., McCormick, A.V. and Eisele, F.L., 2008. Dependence of nucleation rates on sulfuric acid vapor concentration in diverse atmospheric locations. Journal of Geophysical Research: Atmospheres, 113(D10).
Vehkamäki, H., Kulmala, M., Napari, I., Lehtinen, K.E., Timmreck, C., Noppel, M. and Laaksonen, A., 2002. An improved parameterization for sulfuric acid–water nucleation rates for tropospheric and stratospheric conditions. Journal of Geophysical Research:

Atmospheres, 107(D22), pp.AAC-3.

Stier, P., Feichter, J., Kinne, S., Kloster, S., Vignati, E., Wilson, J., Ganzeveld, L., Tegen, I., Werner, M., Balkanski, Y., Schulz, M., Boucher, O., Minikin, A., and Petzold, A.: The aerosol-climate model ECHAM5-HAM, Atmos. Chem. Phys., 5, 1125–1156, https://doi.org/10.5194/acp-5-1125-2005, 2005.

Zhang, K., O'Donnell, D., Kazil, J., Stier, P., Kinne, S., Lohmann, U., Ferrachat, S., Croft, B., Quaas, J., Wan, H., Rast, S., and Feichter, J.: The global aerosol-climate model ECHAM-HAM, version 2: sensitivity to improvements in process representations, Atmos. Chem. Phys., 12, 8911–8949, https://doi.org/10.5194/acp-12-8911-2012, 2012.

---

## Author Response (AR1)

The authors would like to thank all reviewers for the constructive comments. Our replies to all comments are given below, with the original comments in black, and our response in blue. We have revised the manuscript accordingly. All changes made to the manuscript have been marked with Track-Change tool in one of submitted files.

**Executive Editor Comment on gmd-2019-290, Astrid Kerkweg**

Dear authors,

please consider, that the loopuk tables might be changed in the future. Therefore please provide a version number of the tables in the title of the manuscript. Best regards, Astrid Kerkweg (Executive Editor)

Thanks for the suggestion. We have added a version number in the title:

"H2SO4-H2O binary and H2SO4-H2O-NH3 ternary homogeneous and ion-mediated nucleation: Lookup tables version 1.0 for 3-D modeling application"

**Anonymous Referee #1**

This work documents aerosol-nucleation-rate lookup-tables generated based on the results from a kinetic model (Yu et al. 2018ACP). Four major aerosol nucleation mechanisms are considered in these lookup-tables and they cover a wide range of key parameters relevant for aerosol nucleation. The lookup-tables can be used in 3-D models to save computational cost, so they are potentially useful for other modelers who want to simply the aerosol nucleation treatment in their models.

Overall, the manuscript is well written. The documentation is clear and key information is provided.

Thank you for positive comments affirming potential usefulness of the work.

I think the work could be further improved by comparing these tables (calculated nucleation rates) with other widely-used aerosol nucleation parameterizations, so that other users can have an idea about what to expect in their model result. For example, the Vehkamaki et al (2002, hereafter V2002) binary nucleation scheme is used by many aerosol models (e.g. CAM5 Liu et al., 2012GMD, ECHAM5-HAM, Stier et al. 2005ACP, etc). How does the BHN lookup-table compare with V2002? The ioninduced aerosol nucleation is considered in the ECHAM5-HAM2 model (Zhang et al., 2012ACP), using a similar lookup-table method (Kazil and Lovejoy ,2007ChemPhys and Kazil et al, 2010ACP). How does the BIMN lookup-table compare with K2010? In addition, some aerosol models (Wang et al, 2009ACP, Zhang et al, 2012) use the nucleation parameterization for the boundary layer (e.g. Kuang et al., 2008JGR) in combination with the binary nucleation scheme. Can the THN/BIMN/TIMN schemes be used along with such boundary nucleation scheme? It would be nice to provide such information to other users as well.

This a great suggestion. Following the reviewer's advice, we have added to the manuscript a figure and associated discussion (Figure 5, Session 4), comparing nucleation rates calculated based on lookup tables presented in this work and several other aerosol nucleation parameterizations with the CLOUD measurements. We were not able to find in the public domain the lookup-table method based ion-induced nucleation (IIN) of Kazil and Lovejoy (2007) that the referee mentioned. Actually we tried to get the code from the ECHAM5-HAM2 model (Zhang et al., 2012ACP) but were not able to download it as it requires an institution level

license (https://redmine.hammoz.ethz.ch/projects/hammoz/wiki/1\_Licencing\_conditions) which we do not have right now. Instead, we selected a parameterization (Modgil et al. (2005) that was derived from an earlier version of the IIN (Lovejoy et al., 2004). We show in Fig. 5 that there exist large differences in the nucleation rates predicted by different nucleation schemes/parameterizations and the CLOUD measurements provide useful constrain to these nucleation schemes. The lookup tables presented in this work are in the best agreements with CLOUD measurements for all four nucleation pathways, in term of not only the absolute nucleation rates but also the correlation coefficients.

We have pointed out in the revised text that the TIMN scheme can be directly applied to calculate nucleation rates in the whole troposphere (including the boundary layer) and thus one shall not combine BHN/THN/BIMN/TIMN schemes presented in this study with empirical boundary nucleation parameterizations (i.e., EAN and EKN) in regional and global simulations.

Other specific/minor comments: P1L11, abstract: have -> has

Corrected.

P1L12, L30 and throughout the text: Better use either "lon" or "ion".

Done.

P1L25: "for BHN, THN, BIMN, and TIMN" could be deleted

Done.

P2L11: Maybe also mention the nucleation pathways involving organics?

Yes. It is now mentioned.

P4L12: Is RH the hybrid relative humidity or the RH respect to water? Please clarify.

It is the RH respect to water. Clarified in the text describing Table 1.

P4L15-24: It would be nice to provide some quantitative estimate of the lookup-table accuracy here.

Compared to those based on the full model, the deviation of nucleation rates based on the lookup tables is generally within a factor of two, well within the corresponding uncertainty of CLOUD measurements. We added a discussion on this to the text.

P4L24: Can two points for S to get sufficient accuracy?

The dependence of nucleation rates on the surface area, which serves as coagulation sink of prenucleation clusters, is relatively linear and two points for S provide reasonable accuracy (compared to the uncertainties in the model itself and measurements). In the atmosphere, the surface area of pre-existing particles not only serves as coagulation sink but also as condensation sink for H2SO4, and thus has a more profound impact because nucleation rates are highly sensitive to [H2SO4]. For the lookup tables, [H2SO4] is fixed and therefore the dependence of nucleation rates on surface area are relatively weaker. It should be noted that most of existing nucleation parameterizations do not take into the effect of surface area. We have clarified this in the text.

P5L1: extrapolation -> linear?

We use linear extrapolation between Log10J vs surface area. We have clarified this in the text.

P5L21: very lower -> very low

Corrected.

P6L20: the online program (http://apm.asrc.albany.edu/nrc/) didn't work for me (both in safari and firefox). Better fix it before the final publication.

We found that sometime the calculators did not restart automatically when the server was rebooted. This problem has now resolved and the calculators shall be online all time.

**Anonymous Referee #2**

F. Yu and co-workers have developed lookup tables for quickly and efficiently obtaining new-particle formation ("nucleation") rates based on Yu's 2018 model, which includes H2SO4, NH3, H2O and ions. This is a very useful tool for researchers in the atmospheric aerosol field. Since this manuscript doesn't deal with the nucleation model as such (it's taken as a given), I won't comment on any of the potential issues with the model itself, but only on the application described here. However I will note that I fully agree with the executive editor's request of providing version numbers - while Yu 2018 is an impressive model, it is unlikely to be \*perfect\* in the sense that no further improvement would ever be possible.

Thanks for confirming that the lookup tables are very useful for researchers. Agree with regard to the version number. We have added a version number in the title.

Some minor issues to take in to account when preparing the final manuscript: -The authors say that their rates can be compared with typical laboratory measurements. How should wall losses be accounted for in the comparison - is the idea that users should just scale the "S" parameter in the model to roughly fit the losses in the experiment? How well does this actually capture the effect of wall losses (especially in e.g. flow-tube experiments?).

Yes, users can just scale the "S" parameter in the model to roughly fit the losses in the experiment. As long as the air mass in the camber or flow tube is well mixed, it shall reasonably capture the effect of wall losses on cluster formation and nucleation.

-The authors say that Q ranges from 2 to 23 ion pairs / cm3; for the benefit of casual readers just skimming the text they might mention here that the Q=0 case is also covered (as they actually have separate look-up tables for this case).

Good point. We have pointed this out by adding to the sentence "(noting that Q=0 is covered under BHN or THN)".

-"extrapolation is allowed": this sweeping statement sounds potentially a bit dangerous; have the authors actually tested how well extrapolations work? Perhaps give the readers some guidelines on what kind of extrapolations are recommended, and/or some caveats as to when they can be expected to work (and when not)?

Actually, for the code provided in the Zenodo, extrapolation is allowed only for surface area for which the tables only give values at two surface area points (S = 20 and  $200 \ \mu m^2 cm^{-3}$ ). The dependence of nucleation rates on the surface area, which serves as coagulation sink (not condensation sink because [H2SO4] is fixed), is relatively linear and thus extrapolation will not cause unphysical values. We have clarified this in the text.

-There seems to be an extra bracket in TIMN) on line page 3, 24

Corrected.

**Anonymous Referee #3**

This paper describes look-up tables to speed up the implementation of the state-of-theart Yu et al (2018) ternary nucleation parameterization in atmospheric models. These look-up tables have the potential to be very helpful to atmospheric modelers and they are well described both in the manuscript and in the well-documented code available via Zenodo. I recommend this paper for publication. I have only minor improvements to suggest.

Thank you for positive comments about the manuscript and potential usefulness of the lookup tables.

I note that I was able to use the online program successfully, despite the comment of one of the other reviewers, so I can comment that the authors have presumably fixed it.

Yes, we noticed the problem and fixed it.

The parameterization documented by Yu et al (2018) has not been, as far as I am aware, tested under all atmospheric or planetary science conditions, and so I think its range of validity could be discussed in this paper a little more thoroughly to ensure atmospheric modelers are aware of its possible limitations.

It is hard to discuss the range of validity without measurements to compare with. As discussed in Yu et al (2018) and pointed out in the Introduction, the ternary ion-mediated nucleation (TIMN) model for H2SO4-H2O-NH3-Ion system is a kinetic model with thermodynamic data derived from laboratory measurements and quantum-chemical calculations. The model generally agrees well the CLOUD measurements under a range of conditions. The lookup tables are designed to calculate nucleation rates in the troposphere, not under all atmospheric or planetary science conditions. For conditions in the stratosphere (RH<0.5%) and other planets (such as on Venus), we do not know if the model is valid or not as there are not measurements under such conditions are available to validate the model. We slightly extended the discussion on this.

In polluted conditions or where there are high concentrations of biogenic organic molecules, I think it is possible that the HSO4- ion concentrations predicted by the model for a given ion production rate could be biased high since other molecules may be ionized instead. I appreciate that in these conditions the ions will be mostly lost to high condensation sinks, and so nucleation is likely to be dominated by neutral processes. So it is unlikely to be a big effect, but still perhaps worth mentioning.

The initial negative ions assumed in the model is  $NO_3^-$ . While it is possible that other molecules may be ionized instead, these molecules can be replaced by  $HNO_3$  or  $H_2SO_4$  as long as the bonding of negative ions with  $HNO_3$  or  $H_2SO_4$  are stronger. As the reviewer pointed out, small ions will be mostly lost to high condensation sinks or ion-ion recombination, both having already been included in the kinetic nucleation model (Yu et al., 2018).

Similarly, the comment that 'extrapolation is allowed' for conditions out of range of the table might need qualifying, since nucleation rates are very non-linear. While I appreciate that extrapolation from this parameterization should be more robust than extrapolation from, for example, the empirical parameterization of Dunne et al (2016), it still necessarily leads to uncertainty. In particular, I think nucleation rates at very low relative humidity (below 0.5%) or at temperatures above 300K are still very uncertain and extrapolating from the tables may lead to errors. Could the binary part of the parameterization be used for nucleation on Venus, for example, as discussed by Määttanen et al (JGR 2018; https://doi.org/10.1002/2017JD027429), or in the stratosphere?

Actually, for the code provided in the Zenodo, extrapolation is allowed only for surface area for which the tables only give values at two surface area points (S = 20 and  $200 \ \mu m^2 cm^{-3}$ ). The dependence of nucleation rates on the surface area, which serves as coagulation sink (not condensation sink because [H2SO4] is fixed), is relatively linear and thus extrapolation will not cause unphysical values. The lookup tables are designed to calculate nucleation rates in the troposphere, not under conditions in the stratosphere (RH<0.5%) and other planets (such as on Venus) if the conditions are far different from the tropospheric conditions. We have clarified this in the text.

As the authors address the comment of reviewer #1 on comparison to other parameterizations, the Määttanen et al paper should be discussed as it is in some respects an

update of Vehkamaki et al (2002).

Yes, the nucleation rate based on Määttanen et al.'s paper has been included in the comparison suggested by reviewer #1.

On page 3, line 8, there is an extraneous 't'.

Corrected.

On page 6, line 8, I think it's worth

specifying "ternary nucleating systems with ammonia" because while the statement is perfectly correct for the ammonia system, ternary nucleation of other molecules (some amines, for example) with sulfuric acid is dominated by neutral processes.

Modified as suggested.

**H2SO4-H2O binary and H2SO4-H2O-NH3 ternary homogeneous and ion-mediated nucleation: Lookup tables version 1.0 for 3-D modeling application**

Fangqun Yu1, Alexey B. Nadykto1, 2, 3, Gan Luo1, and Jason Herb1

Correspondence to: F. Yu (fyu@albany.edu)

5

1 Atmospheric Sciences Research Center, University at Albany, Albany, New York, US

2 Department of Applied Mathematics, Moscow State University of Technology "Stankin",

Russia

3 National Research Nuclear University MEPhI (Moscow Engineering Physics Institute),

10 Department of General Physics, Russian Federation

Abstract. Formation of new particles in the atmosphere hasve important implications for air quality and climate. Recently, we have developed a kinetically-based H2SO4-H2O-NH3-Ions nucleation model which captures well the absolute values of nucleation rates as well as dependencies of nucleation rates on NH3 and H2SO4 concentrations, ionization rates, temperature, 15 and relative humidity observed in the well-controlled Cosmics Leaving OUtdoor Droplets (CLOUD) measurements. Here we employ the aforementioned recently developed kinetic nucleation model to generate nucleation rate look-up tables for H2SO4-H2O binary homogenous nucleation (BHN), H2SO4-H2O-NH3 ternary homogeneous nucleation (THN), H2SO4-H2O-Ions 20 binary ion-mediated nucleation (BIMN), and H2SO4-H2O-NH3-Ions ternary ion-mediated nucleation (TIMN). A comparison of nucleation rates calculated using the lookup tables with CLOUD measurements of BHN, BIMN, THN, and TIMN is presented. The lookup tables cover a wide range of key parameters controlling binary, ternary and ion-mediated nucleation in the Earth's atmosphere, and are a cost-efficient solution for multi-dimensional modeling. The lookup tables and FORTRAN codes, made available through this work, can be readily used for BHN, 25 THN, BIMN, and TIMN simulations in 3-D modeling. The lookup tables can also be used by

experimentalists involved in laboratory and field measurements for a quick assessment of nucleation involving  $H_2SO_4$ ,  $H_2O$ ,  $NH_3$ , and iHons.

**1. Introduction**

[revised manuscript text omitted]

- Look-up table of steady state nucleation rates for BHN (JBHN), THN (JTHN), BIMN (JBIMN), and 10 TIMN (JTIMN) have been generated under a wide range of atmospheric conditions. There are six parameters controlling JTIMN: sulfuric acid vapor concentration ([H2SO4]), ammonia gas concentration ([NH3]), T, relative humidity (RH), ionization rate (Q), and surface area of preexisting particles (S). Compared to JTIMN, there is one less controlling parameter for both JBIMN (no [NH3] dependence) and JTHN (no Q dependence) while JBHN only depends on four parameters 15 ([H2SO4] T, RH, and S). Table 1 gives the range of each dependent variable dimension, total number of points in each dimension, values at each point, and controlling parameters for the four nucleation pathways. The range and resolution in each parameter space is designed based on the sensitivity of nucleation rates to the parameter, its possible range in the troposphere, and a balance between the accuracy and sizes of the lookup tables. T covers from 190 K to 304 K (resolution: 3 K) and RH (with respect to water) from 0.5% to 99.5% (resolution: 4%). For [H2SO4], we use 31 20 points to cover  $5 \times 10^5$  to  $5 \times 10^8$  cm-3 plus one additional point at [H2SO4] =  $5 \times 10^9$  cm-3. For [NH3], we employ 31 points to cover  $10^8$  to  $10^{11}$  cm-3 plus two additional points at [NH3]=  $10^5$  cm-3 and
- $10^{12}$  cm-3. Q ranges from 2 23 ion-pairs cm-3s-1 with the resolution of 5 values per decade (geometric) plus one additional point at Q =100 ion-pairs cm-3s-1 (noting that Q = 0 ion-pairs cm-3 s-1 is covered under BHN or THN), and S ranges from 20 200 µm2cm-3 with two points. Almost
- all the possible tropospheric conditions relevant to NPF shall be covered with the above parameter ranges. The lookup tables are designed to calculate nucleation rates in the troposphere. For conditions in the stratosphere (RH<0.5%) and other planets (such as on Venus, as discussed in Määttanen et al., 2018), it is unclear if the model is valid or not as measurements under such conditions are not available to validate the model.</li>

The look-up table for JTIMN is the largest, being composed of JTIMN at more than 17 million points (32x33x39x26x8x2=17,132,544) and with the total size in the text format of ~103 MB. For comparison, the smallest lookup table (for JBHN) has just 64,896 points with total size of ~ 0.38 MB in the text format. For any given values of [H2SO4], [NH3], T, RH, Q, and S within the ranges

- 5 specified in Table 1, nucleation rates can be obtained using the look-up tables with an efficient multiple-variable interpolation scheme as described in Yu (2010). For conditions out of the ranges specified in Table 1, which may occur occasionally in the atmosphere, extrapolation is allowed. linear extrapolation is allowed only for surface area for which the tables only give values at two surface area points (S = 20 and 200  $\mu$ m2cm-3). The dependence of nucleation rates on the surface
- 10 area, which serves as coagulation sink (not condensation sink because [H2SO4] is fixed), is relatively linear and thus extrapolation (linearly between Log10J versus surface area) will not cause unphysical values. The JBHN, JTHN, JBIMN, and JTIMN look-up tables can be accessed via the information given in the data availability section and can be used to calculate nucleation rates efficiently in 3-dimensional models.
- 15 Compared to those based on the full model, the deviation of nucleation rates based on the lookup tables is generally within a factor of two, well within the corresponding uncertainty of CLOUD measurements. The dependence of nucleation rates on the surface area is relatively linear and two points for S provide reasonable accuracy (compared to the uncertainties in the model itself and measurements). In the atmosphere, the surface area of pre-existing particles not only serves as
- 20 coagulation sink but also as condensation sink for H2SO4, and thus has a more profound impact because nucleation rates are highly sensitive to [H2SO4]. For the lookup tables, [H2SO4] is fixed and therefore the dependence of nucleation rates on surface area are relatively weaker. It should be noted that most of existing nucleation parameterizations do not take into the effect of surface area.
- 25

**3.** Comparison of BHN, THN, BIMN, and TIMN rates from the lookup tables with CLOUD measurements**

Dunne et al. (2016) reported CLOUD measured nucleation rates under 377 different conditions 30 (Table S1 of Dunne et al. (2016)). These data can be divided into BHN, THN, BIMN, and TIMN based on the values of [NH3] and Q in the chamber. Nucleation is classified as neutral (BHN or THN), when Q=0, and as binary (BHN or BIMN) when [NH3] < 0.1 ppt. As a result, 15, 27, 110, and 225 of these CLOUD measurements correspond to BHN, BIMN, THN, and TIMN, respectively. Figures 1-4 present the comparisons of the nucleation rates calculated from the lookup tables (Jmodel) with corresponding values observed during CLOUD experiments (Jobs) under BHN, BIMN, THN, and TIMN conditions. The error bars give the  $J_{model}$  range as a result of the

5 measured [H2SO4] uncertainty (-50%,  $\pm$ 100%). The uncertainties in Jobs (overall a factor of two)

and those associated with the uncertainty in measured [NH3] (-50%, +100%) are not included. Because of the increase in the contamination (both unwanted ammonia and amines) with the

- CLOUD chamber temperate (Kurten et al., 2016), binary nucleation measurements (i.e., without 10 ammonia, [NH3] < 0.1 ppt) are only available at very lower T (Figs. 1-2). Both BHN and BIMN predictions based on the lookup tables overall agree well with the available CLOUD observations within the uncertainty range. As pointed out earlier, binary H2SO4-H2O clusters co-exist with ternary H2SO4-H2O-NH3 ones in the ternary system while neutral clusters co-exist with charged clusters in the system containing ions. Therefore, the nice agreement of BHN and BIMN model 15 predictions with observations provides good foundation for the more complex THN and TIMN models. CLOUD experiments have more data points for THN and TIMN, under a wide temperature range covering the lower troposphere. For the THN (Fig. 3), the model prediction is consistent with measurements at low temperature (T=  $\sim 205 - 250$  K) but deviates from measurements at high T, with the level of model under-prediction increasing with increasing T. 20 As pointed out in Yu et al., (2018), the level of contamination in the CLOUD chamber appears to increase with temperature (Kurten et al., 2016), the nice agreement at lower T and the deviation at higher T may be associated with contaminations (such as amines, etc.) in the CLOUD (Kirkby et al., 2011) which increase with temperature (Kurten et al., 2016). In contrast to THN,  $J_{model}$  for TIMN (Fig. 4) agrees with CLOUD measurements within the uncertainties under nearly all conditions. Jmodel for TIMN at T=292 - 300 K is slightly lower than the corresponding observed 25 values, likely a result of similar reasons for the THN under-prediction at higher T (Fig. 3). As demonstrated in Yu et al. (2018), the nucleation of ions is typically stronger than that of neutral clusters for both binary and ternary nucleating systems with ammonia. The ubiquitous presence of ionization in the Earth's atmosphere calls for regional and global aerosol models to take into account the effect of ionization in NPF. The BIMN and TIMN lookup tables, derived from a
- 30

physics-based kinetic nucleation model and validated against the state-of-the-art CLOUD

measurements, provide an efficient way to incorporate the role of ionization in new particle formation in 3-D models.

**4. Comparison of BHN, THN, BIMN, and TIMN rates based on the lookup tables with those**

**5 based on other models/parameterizations**

Many global models explicitly calculate nucleation rates but different models/studies employ quite different nucleation schemes (e.g., Wang and Penner, 2009; Yu et al., 2010; Liu et al., 2012; Zhang et al., 2012; Yu et al., 2012; Williamson et al., 2019). For example, BHN scheme of Vehkamaki et al (2002, hereafter V2002) was used by CAM5 (Liu et al., 2012) and ECHAM5-

- 10 HAM (Stier et al. 2005). The H2SO4-H2O ion-induced nucleation (IIN, similar to BIMN defined in this study) of Lovejoy et al. (2004) and Kazil and Lovejoy (2007) was considered in the ECHAM5-HAM2 model (Zhang et al., 2012). The H2SO4-H2O ion-mediated nucleation scheme of Yu and Turco (2001) and Yu (2010) was employed by the GEOS-Chem (Yu et al., 2010) and CAM5 (Yu et al., 2012). In addition, some aerosol models (Wang and Penner, 2009, Zhang et al.,
- 15 2012) used the empirical nucleation parameterization for the boundary layer (e.g. Kuang et al., 2008) in combination with the binary nucleation scheme. It is of interesting to understand the differences in nucleation rates predicted by different nucleation schemes under the well-controlled CLOUD conditions.

Figure 5 compares nucleation rates calculated based on lookup tables presented in this work

- 20 and several other aerosol nucleation parameterizations with the CLOUD measurements. The nucleation models/parameterizations considered in Fig. 5 include this study (i.e., the lookup tables described in this paper), BHN of Kulmala et al. (1998) (K1998) and Vehkamaki et al (2002) (V2002), BHN and BIMN of Yu (2010) (Y2010) and Määttanen et al. (M2018), IIN (same as the BIMN) of Modgil et al. (2005) (M2005) which is a parameterization based on Lovejoy et al.
- 25 (2004), THN of Napari et al. (2002) (N2002), empirical activation nucleation (EAN) parameterization of Riipinen et al. (2007) (EAN-R2007, J=3.5×10-7[H2SO4]), and empirical kinetic nucleation (EKN) parameterization of Kuang et al. (2008) (EKN-H2008, J=2.5×10-13 [H2SO4]2). The EAN and EKN parameterizations were derived from atmospheric nucleation measurements in the boundary layer (at the presence of ammonia and ionization) and thus are compared with the TIMN scheme (Fig. 5d). For THN (Fig. 5c), N2002 scaled by 10-5 has been
- used in some modeling studies (e.g., Williamson et al., 2019) and thus values of N2002  $\times 10^{-5}$  are

also given in Fig. 5c for comparisons. It can be seen from Fig. 5 that there exist large differences in the nucleation rates predicted by different nucleation schemes/parameterizations and the CLOUD measurements provide useful constrain to the nucleation schemes. Among the schemes considered in Fig. 5, the lookup tables presented in this work are in the best agreements with

- 5 CLOUD measurements for all four nucleation pathways, in term of not only the absolute nucleation rates but also the correlation coefficients. BHN rates based on K2008, V2002, and M2018 are generally 1-4 orders of magnitude higher than the observed values, with K1998 has the lowest correlation coefficient (r=0.48). For BIMN, M2005 generally under-predicts while M2008 overestimates the rates by up to ~ 2 orders of magnitude. For THN, N2002 significantly over-
- 10 estimates the rates, by 5-9 orders of magnitude. The scaling of N2002 by 10-5 reduces the overestimation but the correlation coefficient remains low (r=0.32). The empirical parameterizations (both EAN and EKN) depends only on [H2SO4] and, unsurprisingly, have very low correlation coefficients (r=0.08) with CLOUD measurements. Care should be taken in employing the empirical parameterizations in global models as both EAN and EKN may give
- 15 incorrect spatial distributions (Yu et al., 2010) and temporal variations of nucleation rates in the atmosphere. It should be noted that the TIMN scheme can be directly applied to calculate nucleation rates in the whole troposphere (including the boundary layer) and thus one shall not combine BHN/THN/BIMN/TIMN schemes presented in this study with empirical boundary nucleation parameterizations (i.e., EAN and EKN) in regional and global simulations.

20

Acknowledgments. We acknowledge funding support from the US National Science Foundation (grant #: AGS1550816), the Russian Science Foundation, and the Ministry of Science and Education of Russia (grants #: 1.6198.2017/6.7 and 1.7706.2017/8.9).

25

30

**Code and data availability.** The code and lookup tables can be accessed via the zenodo data repository http://doi.org/10.5281/zenodo.3483797. For quick calculation of BHN, THN, BIMN, and TIMN rates under specified conditions, one can use the online nucleation calculators which we have developed based on these lookup tables and made available to the public at http://apm.asrc.albany.edu/nrc/.

**Author Contribution**. FY designed the lookup tables and generated the lookup tables. FY, AN, GL, and JH contributed to the kinetic model used to generate the lookup tables. FY wrote the paper with contributions from all co-authors.

5 **Competing interests**. The authors declare that they have no conflict of interest.

**Table 1**. The range of values for each independent variable in the BHN, THN, BIMN, and TIMN nucleation rate lookup tables. Also given are the total number of values for each variable, the specific values at which nucleation rates have been calculated, and controlling parameters for the four nucleation mechanisms.

| Para  | meters                 | Range               | Total          | Values at each point                           | Controlling Parameters |     |      |      |
|-------|------------------------|---------------------|----------------|------------------------------------------------|------------------------|-----|------|------|
|       |                        |                     | # of
points |                                                | BHN                    | THN | BIMN | TIMN |
| [H    | 2SO4]                  | 5×10 5 – | 32             | $[H_2SO_4](i)=5\times10^5\times10^{(i-1)/10},$ | Х                      | Х   | X    | Х    |
| (0    | $2m^{-3}$ )            | 5×10 9   |                | i = 1, 31                                      |                        |     |      |      |
|       |                        |                     |                | $[H_2SO_4](32)=5\times10^9$                    |                        |     |      |      |
|       | Т                      | 190 - 304           | 39             | $T(j)=190 + 3 \times (j-1), j=1, 39$           | X                      | Х   | X    | Х    |
|       | (K)                    |                     |                |                                                |                        |     |      |      |
| ]     | RH                     | 0.5 - 99.5          | 26             | $RH(1)=0.5, RH(k)=4\times(k-1),$               | х                      | Х   | х    | Х    |
| (     | (%)                    |                     |                | k =2, 25; RH(26)= 99.5                  |                        |     |      |      |
|       | S                      | 20 - 200            | 2              | S(1)= 20, S(2)= 200                            | х                      | Х   | х    | Х    |
| (µm   | $n^2 \text{cm}^{-3}$ ) |                     |                |                                                |                        |     |      |      |
| []    | NH3]                   | $10^5 - 10^{12}$    | 33             | $[NH_3](1)=10^5, [NH_3](m)=$                   |                        | Х   |      | х    |
| (0    | $2m^{-3}$ )            |                     |                | $10^8 \times 10^{(m-1)/10}, m = 2, 32;$        |                        |     |      |      |
|       |                        |                     |                | $[NH_3](33) = 10^{12}$                         |                        |     |      |      |
| Q (io | on-pairs               | 2 - 100             | 8              | $Q(n)=2\times 1.5^{(n-1)}, n=1, 7;$            |                        |     | Х    | х    |
| cn    | $n^{-3}s^{-1}$ )       |                     |                | Q(8)=100                                       |                        |     |      |      |

**Figures**

5 Figure 1. Model predicted (Jmodel) versus observed (Jobs) nucleation rates under BHN conditions (no ionization, [NH3]<0.1 ppt) of CLOUD measurements reported in Table S1 of Dunne et al. (2016). The data points are grouped according to temperatures as specified in the legend. Vertical error bars show the range of Jmodel calculated at 50% and 200% of measured [H2SO4], corresponding to the uncertainties in measured [H2SO4] (-50%, +100%). Error bars associated with the uncertainties in measured [NH3] (-50%, +100%), and Jobs (overall a factor of two) are not shown.

Figure 2. Same as Figure 1 but for BIMN conditions (with ionization, [NH3]